# The challenges of replication: A worked example of methods reproducibility using electronic health record data

Richard Williams[1,2]*, Thomas Bolton[3], David Jenkins[1], Mehrdad A. Mizani[3,4], Matthew Sperrin[1], Cathie Sudlow[3], Angela Wood[3,5,6,7], Adrian Heald[8], Niels Peek[1,9], on behalf of the CVD-COVID-UK/COVID-IMPACT Consortium¶

1 Division of Informatics, Imaging and Data Science, Faculty of Biology, Medicine and Health, University of Manchester, Manchester, United Kingdom, 2 NIHR Applied Research Collaboration Greater Manchester, Manchester Academic Health Science Centre, University of Manchester, Manchester, United Kingdom, 3 British Heart Foundation Data Science Centre, Health Data Research UK, London, United Kingdom, 4 Institute of Health Informatics, University College London, London, United Kingdom, 5 Department of Public Health and Primary Care, British Heart Foundation Cardiovascular Epidemiology Unit, University of Cambridge, Cambridge, United Kingdom, 6 Victor Phillip Dahdaleh Heart and Lung Research Institute, University of Cambridge, Cambridge, United Kingdom, 7 Health Data Research UK Cambridge, Wellcome Genome Campus and University of Cambridge, Cambridge, United Kingdom, 8 Department of Diabetes and Endocrinology, Salford Royal NHS Foundation Trust, Salford, United Kingdom, 9 Department of Public Health and Primary Care, The Healthcare Improvement Studies Institute (THIS Institute), University of Cambridge, Cambridge, United Kingdom

¶ A full list of consortium members is available in the Acknowledgements.
* richard.williams@phc.ox.ac.uk

## Abstract

### Objective

The ability to reproduce the work of others is an essential part of the scientific disciplines. Replicating observational studies using electronic health record (EHR) data can be challenging due to complexities in data access, variations in EHR systems across institutions, and the potential for unaccounted confounding variables. Our aim is to identify the barriers to methods reproducibility for replication studies using EHR data.

### Methods

We replicated a study that examined the risk of hospitalisation following a positive COVID-19 test in individuals with diabetes. Using EHR data from the NHS England's Secure Data Environment (SDE) covering the whole of England, UK (population 57m), we sought to replicate findings from the original study, which used data from Greater Manchester (a large urban region in the UK, population 2.9m). Both analyses were conducted in Trusted Research Environments (TREs) or SDEs, containing linked primary and secondary care data, however methods reproducibility was not straightforward. Differences between the environments that contributed to the

**Data availability statement:** The data used in this study are available in NHS England's SDE service for England, but as restrictions apply they are not publicly available (https://digital.nhs.uk/services/secure-data-environment-service). The CVD-COVID-UK/COVID-IMPACT programme led by the BHF Data Science Centre (https://bhfdatasciencecentre.org) received approval to access data in NHS England's SDE service for England from the Independent Group Advising on the Release of Data (IGARD) (https://digital.nhs.uk/about-nhs-digital/corporate-information-and-documents/independent-group-advising-on-the-release-of-data) via an application made in the Data Access Request Service (DARS) Online system (ref. DARS-NIC-381078-Y9C5K) (https://digital.nhs.uk/services/data-access-request-service-dars/dars-products-and-services). The CVD-COVID-UK/COVID-IMPACT Approvals & Oversight Board (https://bhfdatasciencecentre.org/areas/cvd-covid-uk-covid-impact/) subsequently granted approval to this project to access the data within NHS England's SDE service for England. The de-identified data used in this study were made available to accredited researchers only. Those wishing to gain access to the data should contact bhfdsc@hdruk.ac.uk in the first instance.

**Funding:** The British Heart Foundation Data Science Centre (grant No SP/19/3/34678, awarded to Health Data Research (HDR) UK) funded co-development (with NHS England) of the Secure Data Environment service for England, provision of linked datasets, data access, user software licences, computational usage, and data management and wrangling support, with additional contributions from the HDR UK Data and Connectivity component of the UK Government Chief Scientific Adviser's National Core Studies programme to coordinate national COVID-19 priority research. Consortium partner organisations funded the time of contributing data analysts, biostatisticians, epidemiologists, and clinicians. The associated costs of accessing data in NHS England's secure data environment service for England, for analysts working on this study, were funded by the Data and Connectivity National Core Study, led by Health Data Research UK in partnership with the Office for National Statistics, which is funded by UK Research and Innovation (grant ref:

difficulties were documented, categorized into themes, and converted into a list of recommendations for TRE/SDEs.

## Results

Small differences between the environments and the data sources led to several challenges in methods reproducibility. Our recommendations of TRE/SDEs should facilitate future replication studies. The recommendations include: a need for improved machine-readable metadata for EHR data; standardization of governance processes to facilitate federated analysis; mandating of code sharing; and for environments to have a support structure for data engineers and analysts. We also propose a new theme for research, "data reproducibility", as the ability to prepare, extract and clean data from a different database for a replication study.

## Conclusion

Even with perfect code sharing, data reproducibility remains a challenge. Our recommendations have the potential to reduce the barriers to replication studies and therefore enhance the potential of observational studies using EHR data.

## Introduction

There is a replication 'crisis' in research. A Nature survey of 1,576 researchers found that 52% of respondents thought that there was a significant 'crisis' of reproducibility, and 72% had tried and failed to reproduce someone else's work [1]. A narrative review focusing on health informatics literature found an increasing interest in replication, but a lack of replication studies [2].

There are many reasons for the replication crisis. There is bias towards novelty, with replication studies viewed as second rate when compared with primary studies [1,2]. There is publication bias, where statistically significant results are more likely to be submitted by authors, and more likely to be published by journals [3]. There are also specific reasons related to the domain of observational research with electronic health record (EHR) data. The sensitive nature of healthcare data means there are often barriers to obtaining the same data. When alternative sources of data are used, the collection methods or data structure can often be different, leading to opposite results in many cases [4]. Replication of observational EHR studies, with perhaps more complex methods and data than a typical randomised control trial, can be hampered by high-impact journals' strict word limits, hindering researchers' ability to fully detail their workflows and potentially limit reproducibility.

Replication studies rely on the ability to reproduce several aspects of the original work. Goodman et al. define three terms for discussing research reproducibility: methods reproducibility, results reproducibility and inferential reproducibility [5]. Methods reproducibility is the degree to which a publication includes sufficient information such that other researchers could repeat the analysis. Results reproducibility is the

MC_PC_20058). This research was co-funded by the NIHR Manchester Biomedical Research Centre (NIHR203308) and the NIHR Applied Research Collaboration Greater Manchester (NIHR200174). AMW is supported by the BHF Data Science Centre (HDRUK2023.0239) and as an NIHR Research Professor (NIHR303137). This work was supported by core funding from the: British Heart Foundation (RG/18/13/33946), NIHR Cambridge Biomedical Research Centre (BRC-1215-20014; NIHR203312) [*], Cambridge BHF Centre of Research Excellence (RE/18/1/34212), BHF Chair Award (CH/12/2/29428) and by Health Data Research UK, which is funded by the UK Medical Research Council, Engineering and Physical Sciences Research Council, Economic and Social Research Council, Department of Health and Social Care (England), Chief Scientist Office of the Scottish Government Health and Social Care Directorates, Health and Social Care Research and Development Division (Welsh Government), Public Health Agency (Northern Ireland), British Heart Foundation and Wellcome. The views expressed are those of the author(s) and not necessarily those of the NIHR or the Department of Health and Social Care. The funders had no role in study design, data collection and analysis, decision to publish, or preparation of the manuscript.

**Competing interests:** The authors have declared that no competing interests exist.

degree to which other researchers can achieve the same results using the same or different datasets. Inferential reproducibility is the degree to which different researchers would reach the same conclusion based on similar results.

There are several barriers to methods reproducibility. Authors often describe their methods without providing their data curation or data analysis code. Without this, it is hard to reproduce the methods reliably and with confidence. Even when provided, the code may not be well documented, or may be difficult to implement in a different environment. For retrospective observational studies using EHR data there are further challenges. The raw data has been collected for patient care or for billing purposes, and so must therefore first be cleaned, curated and transformed before it is suitable for analysis. This contrasts with other studies where data are collected primarily for analysis and are structured accordingly. Analysts also have little to no control over the raw data and must make do with what is available which may not be comparable to previous studies. It is therefore important to examine methods reproducibility, and how it can be improved, in retrospective studies using EHR data.

Our team has previously examined the risk factors for hospitalisation following COVID-19 in individuals with diabetes [6] in a regional database. We replicated this study using a national database (England, UK). We had the same data engineers and analysts working, and access to the original code, yet methods reproducibility was not straightforward. Therefore, our objective is to provide a step-by-step breakdown of the methodological and compute-environment differences between the studies, and their implications for researchers working with heterogeneous databases of EHR data. A separate paper provides the actual results of the replication, for this same study, and focuses on the degree to which those results, and inferential reproducibility, can be achieved in a regional vs a national database [7].

---

**Problem**
Despite growing acceptance of the importance of replication studies, they remain rare.

**What is Already Known**
Goodman et al defined the 3 aspects of replication as methods reproducibility, results reproducibility and inferential reproducibility.

**What This Paper Adds**
This study proposes "data reproducibility" as a fourth aspect of replication. While methods reproducibility focusses on replicating the statistical analysis of a study given the same data, "data reproducibility" concerns the ability to prepare, extract and clean data from a different database. This study also provides a set of recommendations for secure data environments to improve all aspects of the replication of scientific studies.

---

## Methods

The original study applied univariable and multivariable analyses to a dataset of individuals with diabetes and matched individuals without diabetes, in order to determine risk factors for hospital admission following a COVID-19 diagnosis [6]. The data was cleaned and transformed into a wide format, with one row per individual, and a series of R scripts produced the outputs [8–10]. For the replication study, once the data was transformed into the same format, replicating the methods was straightforward and the original R scripts were applied unchanged to the new dataset. Therefore, this paper will focus on the dimension of methods reproducibility related to cleaning

and transforming data from a different environment into the format used in our previous study. The statistical methods are therefore not further described but are available in the original paper [6].

This analysis was performed according to a pre-specified analysis plan published on GitHub, along with the phenotyping and analysis code (https://github.com/BHFDSC/CCU040_01).

## Data sources

The original study used data from the Greater Manchester Care Record (GMCR), an integrated shared care record containing primary and secondary care data for the residents of Greater Manchester (population 2.9m).

This study was performed in NHS England's Secure Data Environment (NHS England SDE), which provides access to a range of national data sets relating to healthcare for approved research programmes. In this case, and wherever the NHS England SDE is mentioned in this paper, data were accessed in an instance of the NHSE England SDE made available for COVID-19 research to the BHF Data Science Centre's CVD-COVID-UK/COVID-IMPACT Consortium.

For this study we relied on: primary care data from the General Practice Extraction Service (GPES) Data for Pandemic Planning and Research (GDPPR) [11]; secondary care data from Hospital Episode Statistics (HES) Admitted Patient Care (APC); and COVID-19 test data from the Second-Generation Surveillance System (SGSS) data set which includes almost all community COVID-19 test results in England.

## Analyses

This study consisted of two analyses, each with a cohort comprising individuals with diabetes prior to their first positive COVID-19 test. Diabetes was defined by relevant SNOMED codes in a patient's primary care record. For the first analysis, the positive COVID-19 results were obtained from the primary care record. This was because that was the only source of COVID-19 test results in the GMCR. The first analysis attempted to reproduce the data to be as similar as possible to the original. The second analysis used COVID-19 test and diagnosis data from the primary care record, and test data from the SGSS. This is summarised in Table 1.

Differences between the GMCR and the NHS England SDE were recorded during the execution of the replication study. On completion, we collaborated with the coordinating team of the CVD-COVID-UK/COVID-IMPACT consortium (authors TB, MM, AJ and CS) to ensure a balanced view of both environments.

## Ethics

The North East – Newcastle and North Tyneside 2 research ethics committee provided ethical approval for the CVD-COVID-UK/COVID-IMPACT research programme (REC No 20/NE/0161) to access, within secure trusted research

**Table 1. Differences between the original GMCR study and the two analyses in this replication study.**

|  | Original GMCR study | This study – 1st analysis | This study – 2nd analysis |
|---|---|---|---|
| **Population** | Patients registered with a GP in Greater Manchester. Does not include individuals who have opted out of secondary use of their GP data. | Patients registered with a GP in England, UK, in practices that opted-in for GPES extraction*. Does not include individuals who have opted out of secondary use of their GP data. | |
| **Primary care data** | Direct feed from GP practices. Containing all events in the patient record. | Data from the GDPPR dataset. Contains a subset of records in the patient record that were both available via GPES and considered relevant to pandemic planning and research. | |
| **Admission data** | Direct feed from each hospital within GM | HES APC data | |
| **COVID-19 test data** | From GP record | From GP record | From SGSS data and GP record |

*98% of practices in England

environments, unconsented, whole-population, de-identified data from electronic health records collected as part of patients' routine healthcare.

### Themes

The results are structured around the following themes inspired by the Goldacre review into the safe use of data for health research [12]: access and governance; the research environment; data feeds; and data management, curation and sharing.

## Results

### Access and governance

**How researchers obtain access.** In the GMCR, researchers complete an application outlining their research question; data requirements; and statistical methods. Approval is via the Secondary Uses and Research Governance group (SURG) who ensure that the data requested is proportionate to the research question, and that the statistical approach is suitable.

For our original study, the legal basis for processing the patient data for COVID-19 research was via a notice from the UK Secretary of State for Health under section 5b of The Health Service (Control of Patient Information) Regulations 2002. Following the expiry of the notice in 2022, we obtained approval from the UK Health Research Authority to conduct any health research and are no longer restricted to COVID-19 studies.

For access to data in the NHS England SDE, the CVD-COVID-UK/COVID-IMPACT research programme received approval to access data from the Independent Group Advising on the Release of Data via an application made in NHS England's Data Access Request Service (DARS) Online system (ref. DARS-NIC-381078-Y9C5K). Furthermore, the North East – Newcastle and North Tyneside 2 research ethics committee provided ethical approval for the CVD-COVID-UK/COVID-IMPACT research programme (REC No 20/NE/0161) to access, within secure trusted research environments (TRE), unconsented, whole-population, de-identified data from EHRs collected as part of patients' routine healthcare. The CVD-COVID-UK/COVID-IMPACT Approvals & Oversight Board, comprising representatives from data custodians, data controllers, researchers and public contributors, reviews all project proposals to ensure that they fall within the scope of the regulatory and ethical approvals.

Researchers request access, with justification, to the datasets required for their project. Following approval, accredited analysts are granted access to the NHS England SDE. The analysts' institution (here, University of Manchester) must be named as a joint data controller in the data sharing agreement (DSA) with NHS England. Project leads and their named analysts are also expected to undertake their research in line with the CVD-COVID-UK/COVID-IMPACT consortium's "ways of working" document, which details how projects should be run, promotes cross-institutional collaboration and requires all outputs to be published in open-access publications, and for the analysis code (usually a combination of R, PySpark, and SQL scripts), code lists, phenotyping algorithms and protocol to be made publicly available in a GitHub repository.

**Data discovery.** Analysts need to know in advance of an application whether the data source is likely to contain the necessary information to enable their project. Metadata catalogues are a standard way to present this information. In addition, both environments have an iterative process where applications are reviewed by people with expertise in the underlying data. For the NHS England SDE this is the BHF DSC Health Data Science Team, and for the GMCR it is a group of research data engineers (RDEs). The feasibility of projects is assessed, recommendations for alterations are suggested, and advice is provided on the most suitable datasets and data items.

**Publications.** In the GMCR, publications are checked by SURG in order to ensure the validity and credibility of the results and to preserve the reputation of the GMCR from substandard research. In the NHS England SDE, publications

undergo an initial check by the BHF Data Science Centre's coordinating team for any factual inaccuracies, to ensure that the project's outcomes remain within scope of the CVD-COVID-UK/COVID-IMPACT research programme's regulatory and ethical approval, the appropriate acknowledgment statements have been included, and the required content has been uploaded to the project's GitHub repository. Draft manuscripts are then shared with all members of the CVD-COVID-UK/COVID-IMPACT Consortium for peer review. The BHF Data Science Centre also provides opportunities for Patient and Public Involvement and Engagement (PPIE).

The difference here is that while both environments would block publications, at the time of the study, only the GMCR had documentation detailing how and why this might occur. The writing of this publication highlighted the inconsistency and the next version of the CVD-COVID-UK/COVID-IMPACT ways of working document was updated accordingly.

## Research environment

**Technology.** In the GMCR, data are stored in a Microsoft SQL Server database with column-store indexes, transformed with SQL, and made available to analysts via a secure file share system. The study teams access the environment via a remote virtual desktop environment running Windows, and analyse the data using R or Stata. Users have database read-only access and cannot create permanent database tables.

In the NHS England SDE, data are stored in column-oriented tables (Delta tables in Amazon Web Services (AWS)) and accessed through Hive metastore in a Databricks analytics platform, Apache Spark, RStudio Pro IDE, RStudio Server, or AWS virtual desktop solution for Stata. Analysts log into the NHS England SDE via a portal to access a Windows-based Virtual Desktop Interface using supported browsers and two-factor authentication. The data are also read-only in this environment, but analysts can create tables in a collaboration database with both read and write permissions.

**Import control.** The GMCR and the NHS England SDE did not enforce import checking at the time of the studies. It was possible to copy code snippets, scripts and small text-based reference data such as code lists directly into the environment. The NHS England SDE now enforces input checking for files up to 1MB, with larger files via a special request.

**Export control.** In the GMCR, all results and aggregate data must first be checked by another analyst for disclosure risk prior to exporting. However, there is no technical mechanism to enforce this, relying instead on user training. The next version will have an independent output checking process that is technically enforced.

The NHS England SDE operates a Safe Output Service to maintain disclosure control rules in aggregated results, and so no data elements are visible in exported code. The independent output checking team ensure that aggregate/summary-level results are appropriately disclosure controlled and justified by supporting contextual information.

**Execution time.** In the GMCR, queries run relatively quickly, due to the smaller population. In the NHS England SDE, the larger population (57m vs 2.9m), means some queries take longer to run. However, analysts can save intermediate database tables, allowing them to run time-consuming queries once, before caching the results for future use.

For analysts attempting to repeat analyses with larger datasets this can potentially be an obstacle. Computationally intensive statistical or machine learning methods, for example multiple imputation or bootstrapping, may run quickly in one environment, but take an unreasonable amount of time, or fail in another.

## Data feeds

The GMCR and the NHS England SDE, at the time of each study, were databases containing linked primary and secondary care data for the purpose of COVID-19 research.

**GP data.** In the GMCR, GP data comes from a direct feed from each practice. It includes the entire medical history of clinical codes (currently SNOMED, but at the time of the study Readv2 and CTV3) for each patient. The NHS England SDE uses GP data from the GDPPR dataset [11] which includes over 36,400 SNOMED concepts. This represents a substantial amount of patient data as they are typically the concepts most frequently used by GPs. However there are

SNOMED concepts used by GPs that do not appear in GDPPR. Individuals who opt out of data sharing for secondary use are not included in either database.

The GDPPR does not include testosterone level, vitamin D level, and sex hormone binding globulin (SHBG) level, so these covariates in the original study could not be used. It is also missing some of the severe mental illness codes used in the original study, particularly symptom codes. Our original plan was to replicate a study focusing on COVID-19 and mental health, but this was not possible due to the reduced set of diagnoses available in GDPPR.

**COVID-19 tests.** The GMCR lacks a dedicated COVID-19 test database, so tests are taken from the GP record. In the NHS England SDE, COVID-19 tests are available from several sources (in addition to those in GDPPR):

• Second Generation Surveillance System (SGSS) includes first positive pillar 1 and pillar 2 tests

• Pillar 2 Antigen (positive and negative)

• Pillar 3 Antibody (positive and negative)

• COVID-19 Hospitalisation in England Surveillance System (CHESS)

• HES data (though this contains diagnoses of COVID rather than actual test results)

**Hospital admissions.** The NHS England SDE has access to HES and SUS data including admitted patient care (APC) data for hospital admissions. In the GMCR, admissions came from direct data feeds from each hospital. However, some hospital feeds did not start until May 2020, and historic data was not available.

**Date of death.** In GMCR this is redacted to month of death. In the NHS England SDE this is not redacted. Date of death comes from the Civil Registrations of Death dataset which is not considered potentially identifiable in the NHS metadata catalogue. GDPPR also provides date of death, which is considered potentially identifiable for this dataset, highlighting a lack of consistency.

In the GMCR, we could not calculate the commonly used outcome of "death within 28 days of a positive COVID-19 test" because we only had access to individual's month of death. We needed to work with the system supplier (Graph-Net Health Ltd) to add a field to the database with this information which could be calculated before the data was pseudonymised.

## Data management, curation and sharing

**Data curation.** The GMCR distinguishes between data curation and data analysis activities. Data curation involves processing, transforming and cleaning the data, by a small team of research data engineers (RDEs) with expertise in EHR data, software engineering, database management and querying. Study teams submit proposals explaining their research questions and the data required. RDEs assess feasibility and suggest alterations. The RDEs provide data to the analysts, minimised to that required to answer the research questions, cleaned and ready for loading into statistical software. Data analysis is performed by the study teams.

In the NHS England SDE, analysts can undertake both data curation and data analysis. The BHF Data Science Centre Health Data Science Team provide different levels of data curation and analysis support to projects and analyst teams including signposting to resources, providing data curation guidance, performing exploratory data analysis, developing and reviewing data curation pipelines. Inductions, further technical support, and help with resolving data queries is provided by the NHS England Data Wrangler Team. This means that support can be tailored to analyst teams with varying levels of experience and also targeted to where support is most needed to speed research productivity.

**Data management and sharing.** The GMCR uses a custom SQL templating language to create extract scripts which are then compiled into raw SQL. This ensures consistent use of common chunks of reusable SQL, and lists of clinical codes, across multiple projects without error. The compiled SQL can then be copied from the RDE's local machine to the

secure VDE and executed against the database to produce flat files, usually csv, which are provided to the analysts. The RDEs also build and maintain a public library of clinical code sets, phenotypes and reusable database queries (https://github.com/rw251/gm-idcr).

The data extraction code for each project within the GMCR is publicly available at the GitHub repository. During compilation, any clinical code sets are automatically collated into a single csv file, and any metadata related to the various chunks of SQL are also automatically extracted and made available in a single README file. This is the README file for the original GMCR study described in this paper (https://github.com/rw251/gm-idcr/blob/master/projects/020%20-%20Heald/README.md).

In the NHS England SDE, data manipulation and curation are performed using Databricks notebooks, collaboration tools, version control through an internal GitLab, and analytics capabilities. Code can be developed directly within the notebooks, as stand-alone Python files, or in an IDE with GitLab integration. Code snippets and text can be copy/pasted into the SDE.

The data curation and analysis pipelines, in addition to dataset summaries and exploratory analyses, are shared via collaboration workspaces for analysts to reuse/adapt to help reduce the amount of time spent on data preparation and to accelerate research. Analysts can also make use of existing code and tables developed by others. All related analysis plans, protocols, code, phenotype code lists and reports are made publicly available via the BHF Data Science Centre's collection on the HDR UK Gateway and HDR UK Phenotype Library, repositories in the BHF Data Science Centre's GitHub organisation, and through open-access publications.

## Discussion

### Summary

This paper compares the two secure data environments (SDEs) used to replicate the results of a regional observational study of EHR data in a national database. We have shown that methods reproducibility is hard even with perfect sharing of the study definition, algorithms and clinical code sets. Differences in the data, and in the environments themselves, are a barrier to quick replication of existing studies.

We will now reflect on the implications for researchers and provide a series of recommendations for Trusted Research Environments (TREs) and SDEs to improve the ease with which replication studies can be performed. The full list of recommendations is in Table 2. These recommendations are relevant to the UK NHS sub-national and regional SDE programme, launched in 2022, which aims to create a network of SDEs across England [13]. These environments, developed through NHS-university partnerships, give researchers controlled access to anonymized NHS patient data.

Another relevant initiative is the UK TRE community's set of standards for best practice for TRE/SDEs. This includes the standardised architecture for trusted research environments (SATRE) [14], a federated network of TREs (TRE-FX) [15], and software for the semi-automated checking of research outputs (SACRO) [16].

### Access and governance

Different access and governance arrangements can hinder replication. Most SDE programmes aim for researchers to be able to federate their analysis across multiple environments. However, separate application forms and different governance arrangements will add a considerable burden to researchers, and likely mean federation would never happen in more than a couple of environments.

**Recommendation 1:** SDEs/TREs that wish to allow federated analysis should consider unified application processes so that researchers need only apply once.

The GMCR governance arrangements mean that sub-standard research can be blocked from publication in the interest of preserving its reputation. This is also true for the SAIL databank [17] where research in breach of their output review

**Table 2. Full list of recommendations for Trusted Research Environments (TREs) and Secure Data Environments (SDEs) to facilitate replication studies.**

| Recommendation 1 | SDEs/TREs that wish to allow federated analysis should consider unified application processes so that researchers need only apply once. |
|---|---|
| Recommendation 2 | SDEs/TREs should clearly define permissible research outputs, the assessment process, and any appeals procedures. |
| Recommendation 3 | Metadata catalogues designed specifically for longitudinal EHR data should be researched and developed. |
| Recommendation 4 | Research is needed to develop computable study definitions that can be executed against machine-readable metadata catalogues |
| Recommendation 5 | SDEs/TREs should consider how automation and other efficiencies can reduce access costs. Ensuring that replication and federated analyses remain affordable is crucial for the advancement of research. |
| Recommendation 6 | SDEs/TREs should be designed to be agile and adaptable, incorporating best practices as they evolve. The SATRE specification is the most likely source for these best practices. |
| Recommendation 7 | SDEs/TREs should ensure that data can be accessed and processed with multiple languages such as SQL, R and Python. |
| Recommendation 8 | SDEs/TREs should implement mechanisms to monitor and manage execution time variability. Providing researchers with tools to estimate and optimise execution times can improve the efficiency and reliability of data analysis. |
| Recommendation 9 | SDEs/TREs should allow safe content to be easily imported. Where import controls are enforced, they should minimize the barriers to researchers. |
| Recommendation 10 | SDEs/TREs need a support structure for researchers which includes people with expertise in the underlying data. |
| Recommendation 11 | Libraries of code, clinical code sets and phenotypes should consider their editorial policy. If there are no barriers to uploading content, then standardised tools should be created to allow easy discovery and comparison of the digital artifacts. |
| Recommendation 12 | SDEs/TREs should mandate the sharing of data curation and data analysis code. |

policy can be blocked. For example, performance tracking of individual organisations is not permitted. Research carried out in the NHS England SDE is subject to statistical disclosure control, subject to checks by the BHF Data Science Centre and then subject to peer review by the CVD-COVID-UK/COVID-IMPACT consortium. To avoid replication issues where a study is possible in one environment but blocked from publication in another we recommend:

**Recommendation 2:** SDEs/TREs should clearly define permissible research outputs, the assessment process, and any appeals procedures.

**Metadata.** In the GMCR, we found that providing researchers with database field descriptions is unhelpful because additional information is missing. Typical metadata for a field, such as the name and description, lacks measures of completeness, how usage varies over time, and whether information is redacted. Also, the complexity of EHR data is restricted to the fields containing clinical codes, which contain all medical concepts from diagnoses and procedures to medications and results. This contrasts with data from randomised controlled trials or cohort studies where each measurement or observation will have a separate field. Standard metadata catalogues are designed for these controlled studies but are inappropriate for longitudinal EHR data as demonstrated previously [18]. Also, the data provided to analysts is transformed into a research-ready format and may bear little relationship to the underlying database structure, so it is better to describe the available data in broad terms and offer a service where preliminary ideas can be checked for feasibility.

This highlights a need for improved metadata catalogues specifically designed for EHR data. However, several of the discrepancies encountered in the data in the two environments, such as how a hospital admission is defined, or which clinical codes are available, would likely not be documented in a data catalogue. Even if they were, the large volume of information in the metadata catalogue could reduce its utility. Computable study definitions, combined with machine-readable metadata catalogues, might enable feasibility checking and automatic execution of replication studies. However,

given the difficulty of finding two datasets with all variables necessary for a particular study to be recorded in the same way, and given that these differences can make data reproducibility problematic, it may be some time before we can achieve this, even when utilising data within one national digital health infrastructure.

**Recommendation 3:** Metadata catalogues designed specifically for longitudinal EHR data should be researched and developed.

**Recommendation 4:** Research is needed to develop computable study definitions that can be executed against machine-readable metadata catalogues.

**Access costs.** SDEs are the current standard for conducting safe research, with the sharing of code and tools in these environments expected to lead to an acceleration of research [12]. However, research conducted in an SDE can be more expensive than the traditional way of distributing copies of the data, due to the additional technical and administrative costs passed onto researchers via an access fee. Previously the costs of storage and compute using local university resources may have been hidden from the study teams. Large access fees could reduce the number of research groups conducting research in these environments, potentially limiting the benefits of SDEs to research quality and safety, rather than increasing quantity. Federation adds to this problem, where the cost would be prohibitive if a large fee was required for each SDE.

**Recommendation 5:** SDEs/TREs should consider how automation and other efficiencies can reduce access costs. Ensuring that replication and federated analyses remain affordable is crucial for the advancement of research.

## Research environment

**Environment heterogeneity.** Small differences between environments can significantly impact users' interactions with them. These differences are unlikely to become apparent until after accessing the environment. One example from this study, is the ability to create permanent database tables, which was possible in the NHS England SDE but not in the GMCR. Therefore, in the GMCR, all interim calculations were done via temporary tables. These tables only last for the lifetime of the query and so data caching for improved performance on subsequent queries is not possible. Queries must also be deterministic and not random. For example, where a matched cohort is required in multiple queries the ideal situation would be to define the cohort once, save it to a permanent table, and then use it in subsequent queries. The limitation means that instead the cohort must be created in every query that it is used in, and it must generate the exact same matched cohort each time. This is a seemingly trivial difference, and one that would be unknowable until access had been granted, but it has a significant effect on the interactions with the system.

**Recommendation 6:** SDEs/TREs should be designed to be agile and adaptable, incorporating best practices as they evolve. The SATRE specification [14] is the most likely source for these best practices.

**Execution time.** An example of the replication issue where methods do not scale to a larger cohort is for the matching that is required for cohort and case-control studies. In the GMCR, matching on sex, age and date of COVID-19 test is done via a loop in SQL, which progressively relaxes the matching criteria for individuals without matches. This approach scaled polynomially and was an acceptable solution for GM, but consumed too much memory when applied to the national data. The cohort matching was rewritten in Python and the algorithm improved by pre-sorting the data. The relevant data was extracted and loaded into Pandas data frames, the matching performed, and then the results written back to the database. This improved performance sufficiently, but required significant development time.

**Recommendation 7:** SDEs/TREs should ensure that data can be accessed and processed with multiple languages such as SQL, R and Python.

**Recommendation 8:** SDEs/TREs should implement mechanisms to monitor and manage execution time variability. Providing researchers with tools to estimate and optimise execution times can improve the efficiency and reliability of data analysis.

**Import controls.** Neither environment previously had checks on imported content like clinical code sets or analysis code below a certain size. This leads to a good experience for analysts who can simply copy/paste content into the environment. The SATRE specification [14] has an optional requirement for an approval process (ref 2.1.13), but this might not be justified. While it could prevent the malicious or accidental compromising of the SDE, a malicious user could simply write their malicious code within the environment, albeit more slowly. Instead, future SDE/TREs should be safely sandboxed to make malicious code ineffective. Combined with statistical disclosure control for outputs, this would lead to the best user experience while preserving the integrity of the environment. In any event, an essential requirement for replication studies is for existing code to be imported.

**Recommendation 9:** SDEs/TREs should allow safe content to be easily imported. Where import controls are enforced, they should minimize the barriers to researchers.

## Data feeds

Both environments in this study contained "linked primary and secondary care data from the UK". The assumption would be that studies requiring this sort of data would be possible in either environment. However, hidden differences emerged during data exploration. For example, certain SNOMED codes related to severe mental health were unavailable in GDPPR which prevented replication of a separate GMCR study. Therefore, anyone attempting to replicate from a local database with the full GP record, must evaluate the availability of equivalent or proxy variables in the datasets that are limited extracts of GP records, such as GDPPR.

Working with EHR data requires making the most of what you have as changes to data collection is not possible. The most important data items for these studies were: diabetes diagnosis, positive COVID-19 tests, and hospital admission. While the two databases in this study have similar data and a common purpose, these key data items differ. In the GMCR, COVID-19 tests and diabetes diagnoses were taken from the GP record, and hospital admissions were from direct hospital feeds (i.e., not HES data). In the NHS England SDE, COVID-19 tests were from the SGSS and GP data, diabetes diagnoses were from GDPPR, and hospital admissions were from HES APC data. These differences may or may not affect the results of the replication, but they emphasize the need for sufficiently detailed metadata to ensure these differences are explicit as they will undoubtedly affect methods reproducibility.

## Data management, curation and sharing

Analysing EHR data involves 3 distinct phases: familiarity, engineering (or curation) and analysis. Data familiarity is the understanding of the provenance of the underlying data and its structure. Data engineering/curation is transforming the raw data into a format ready for analysis. Data analysis is the application of statistical methods to the transformed data to produce results.

Data engineering and data analysis are separate domains, with different tools, languages and skillsets, and it is better to have people with expertise in each rather than both. If environments require individuals to have expertise in both, then research is slowed down due to the lack of such people. Separating these activities makes it is easier to find experts in each area, lowering the barrier to research. While engineering and analysis skills can be brought to a new environment, data familiarity must be developed for each new environment.

The RDE model in the GMCR speeds up research by removing the time required for external researchers to develop data familiarity. Instead, a small team of engineers, with an in-depth understanding of the data and engineering skills, provide data analysts with bespoke datasets for their analysis. Therefore, researchers using the GMCR do not need data

familiarity or data engineering skills. Researchers without these skills may struggle when moving to an environment without RDE support.

**Recommendation 10:** SDEs/TREs need a support structure for researchers which includes people with expertise in the underlying data.

The reuse of data wrangling code, clinical code sets and phenotypes within the GMCR is completely managed by the RDEs who have full editorial control. In the national database, at the time of the study, these digital artefacts were stored within each project directory. Reuse is encouraged, but it can sometimes be hard to find the relevant cleaning or analysis code. When code is found, it can be hard to select between multiple similar options. This problem also occurs on sites that help users share their clinical code sets such as clinicalcodes.org [19] or the HDRUK gateway [20]. By making the sharing of clinical code sets easy, there is a proliferation of similar code sets, particularly for common long-term conditions.

**Recommendation 11:** Libraries of code, clinical code sets and phenotypes should consider their editorial policy. If there are no barriers to uploading content, then standardised tools should be created to allow easy discovery and comparison of the digital artifacts.

Data preparation code is shared automatically in the GMCR, but data analysis code is only shared if the study team choose to do so – it is not mandated. The NHS England SDE through the BHF Data Science Centre mandates that all preparation and analysis code is shared. However, simply putting something on GitHub does not necessarily mean that it is easily reused. SDEs should consider mandating the sharing of analysis code, with a focus on enhancing reusability, for example by using RO-Crates [21,22].

**Recommendation 12:** SDEs/TREs should mandate the sharing of data curation and data analysis code.

## Data reproducibility

Expanding on Goodman's three types of reproducibility [5], we propose a fourth: data reproducibility. While methods reproducibility focusses on replicating the statistical analysis of a study given the same data, "data reproducibility" concerns the ability to prepare, extract and clean data from a different database. In retrospective observational research, an author can provide all of their code, well documented, but if the person attempting replication is using data from a different source, then there is still a data transformation and cleaning exercise required which will affect the reproducibility. In this case the methods reproducibility would be simple, but the data reproducibility would remain hard.

In this study, methods reproducibility was trivial after transforming the data into the format required by the R scripts from the original study. When applied to data in the same format, but from the NHS England SDE rather than the GMCR, the code ran without exception. One minor change was needed because GMCR admissions data are stored in a way such that patients who have not been discharged have a blank discharge date. In the HES APC data in the NHS England SDE, the discharge date field is never blank and an ancient date such as 1800-01-01 indicates an undischarged patient. This was spotted at the point of analysis as a handful of patients had very large negative lengths of hospital stay. While this could have been amended in the data curation code, we instead improved the analysis code to correctly handle records with a negative length of hospital stay. This was noticed easily because the large negative values skewed the results significantly. However, one limitation of replication studies is that it is possible that similar data changes could introduce mistakes that were not as easily detectable.

A related study showed that while a common protocol for studies is helpful, it is not sufficient to remove all the bias of using different databases [23]. Madigan et al [4] found that database choice can influence findings, with 36% of the 53 drug/outcome pairs that were analysed had statistically significant decreased risk in some databases, but statistically significant increased risk in others.

## Other environments and models

Our review has focused on two secure data environments containing linked primary and secondary data. There are several other such environments. OpenSAFELY [24] ensures that researchers do not access the data directly. Queries are constructed outside the environment, executed within the environment, and then the results are presented back to the researchers. OHDSI [25] require participating centres to transform their data into the OMOP common data model. This then allows researchers to execute code against multiple centres and collate the results. CPRD [26] and the SAIL Databank [17] currently implement TREs to allow researchers to access healthcare data in a secure environment.

We believe that there would be similar issues when attempting to replicate between any of the environments described above and that our recommendations would still apply.

## Limitations

The findings are specific to the UK's healthcare data systems which potentially limit the paper's applicability to countries with different healthcare data practices. Further research could explore similar replication studies in other healthcare systems to enhance the generalisability of the recommendations. However, although the recommendations were developed using UK health data, they are presented in a generalised way that will very likely apply in other contexts. Recommendation 3 is only applicable to countries with longitudinal healthcare data, and recommendations 2 and 9 are only applicable in domains where import/export controls are required. The remainder will almost certainly be relevant anywhere that sensitive data is managed in a secure environment.

## Conclusion

By conducting a replication study, we have demonstrated that methods reproducibility faces major difficulties even with perfect sharing of code. It is straightforward to share the cleaned data definition, and the statistical code used to analyse it. However, data reproducibility remains challenging. Our recommendations, together with future research on making study definitions and metadata catalogues machine-readable, should reduce the barriers to replication studies, and elevate the potential of observational studies using EHR data. This is particularly relevant at a time when electronic health record data are increasingly being used to guide national and international health policy direction.

## Supporting information

**S1 File. Full list of members of the COVID-IMPACT consortium.**
(PDF)

## Acknowledgments

This study makes use of de-identified data held in NHS England's SDE for England, and made available via the BHF Data Science Centre's CVD-COVID-UK/COVID-IMPACT consortium. This work uses data provided by patients and collected by the NHS as part of their care and support. We would also like to acknowledge all data providers who make health relevant data available for research.

The full list of members of the COVID-IMPACT consortium can be found in the Supporting Information file.

## Author contributions

**Conceptualization:** Richard Williams, Angela Wood.

**Data curation:** Richard Williams.

**Investigation:** Richard Williams.

**Methodology:** Richard Williams, Thomas Bolton, David Jenkins, Mehrdad A Mizani, Matthew Sperrin, Cathie Sudlow, Angela Wood, Adrian Heald, Niels Peek.

**Supervision:** Niels Peek.

**Writing – original draft:** Richard Williams.

**Writing – review & editing:** Richard Williams, Thomas Bolton, David Jenkins, Mehrdad A Mizani, Matthew Sperrin, Cathie Sudlow, Angela Wood, Adrian Heald, Niels Peek.

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
