## [Decision Letter · Decision Letter 0]

PONE-D-24-52964

The challenges of replication: a worked example of methods reproducibility using electronic health record data

PLOS ONE Dear Dr. Williams,

Thank you for submitting your manuscript to PLOS ONE. After careful consideration, we feel that it has merit but does not fully meet PLOS ONE’s publication criteria as it currently stands. Therefore, we invite you to submit a revised version of the manuscript that addresses the points raised during the review process.

We look forward to receiving your revised manuscript.

Kind regards,

Youssef El Khatib, Ph.D.

Academic Editor

PLOS ONE

Journal Requirements:

[This work is carried out with the support of the BHF Data Science Centre led by HDR UK (BHF Grant no. SP/19/3/34678).]

[The author(s) received no specific funding for this work.]

3. For studies involving third-party data, we encourage authors to share any data specific to their analyses that they can legally distribute. PLOS recognizes, however, that authors may be using third-party data they do not have the rights to share. When third-party data cannot be publicly shared, authors must provide all information necessary for interested researchers to apply to gain access to the data. (https://journals.plos.org/plosone/s/data-availability#loc-acceptable-data-access-restrictions)

4. One of the noted authors is a group or consortium [CVD-COVIDUK/COVID-IMPACT]. In addition to naming the author group, please list the individual authors and affiliations within this group in the acknowledgments section of your manuscript. Please also indicate clearly a lead author for this group along with a contact email address.

Additional Editor Comments:

The paper is of significant value and can be considered for publication pending minor revisions in accordance with the reviewer's suggestions.

Reviewers' comments:

Reviewer's Responses to Questions

**Comments to the Author**

1. Is the manuscript technically sound, and do the data support the conclusions?

Reviewer #1: Partly

2. Has the statistical analysis been performed appropriately and rigorously? 

Reviewer #1: N/A

3. Have the authors made all data underlying the findings in their manuscript fully available?

Reviewer #1: No

4. Is the manuscript presented in an intelligible fashion and written in standard English?

Reviewer #1: Yes

5. Review Comments to the Author

Reviewer #1: I congratulate the authors on their interesting paper on a working example of a reproducibility study with electronic health record (EHR) data.

I found the paper very well written and IMO the paper discusses very important challenges and possible solutions, especially for the problem of data reproducibility. In the worked example, the challenge of reproducing the results is not in the data analysis or statistical analysis part, as all the scripts relevant for these analyses are available and applicable, provided that the data tables are set up exactly as in the study for which a replication study is planned. The challenge is to obtain the data in the same structure from different databases. The conclusions are summarised in 12 recommendations for so-called Trusted Research Environments (TREs) and Secure Data Environments (SDEs) to manage such replication studies.

A drawback of the paper is certainly that the example study presented is very much tailored to how the UK system (NHS) manages access to EHR data. However, I think that the recommendations can be applied in a similar way to health systems in many countries, i.e. I suspect that there is also value in reading about the challenges for research groups not based in the UK.

I suggest that the authors write a paragraph on how they think their 12 recommendations can be generalised to other health systems (although they have described this problem in a limitation paragraph). This would add value to the paper.

A second drawback is that the data are not publicly available, but the authors have done some effort to describe the necessary procedures. I think it is an open question whether other researchers from countries other than the UK will be able to get access to the data since only accredited researchres were allowed to have access.

6. PLOS authors have the option to publish the peer review history of their article (what does this mean? ). If published, this will include your full peer review and any attached files.

**Do you want your identity to be public for this peer review?** For information about this choice, including consent withdrawal, please see our Privacy Policy .

Reviewer #1: No

---

## [Author Response · Author response to Decision Letter 1]

12 May 2025

Please see the "response to reviewers" document.

---

## [Editor Report · Decision Letter 1]

The challenges of replication: a worked example of methods reproducibility using electronic health record data

PONE-D-24-52964R1

Dear Dr. Richard Williams,

We’re pleased to inform you that your manuscript has been judged scientifically suitable for publication and will be formally accepted for publication once it meets all outstanding technical requirements.

Kind regards,

Youssef El Khatib, Ph.D.

Academic Editor

PLOS ONE

Additional Editor Comments (optional):

In my view, the revised version of the paper is suitable for publication.
---

## [Editor Report · Acceptance letter]

PONE-D-24-52964R1

PLOS ONE

Dear Dr. Williams,

I'm pleased to inform you that your manuscript has been deemed suitable for publication in PLOS ONE. Congratulations! Your manuscript is now being handed over to our production team.

Kind regards,

on behalf of

Prof. Youssef El Khatib

Academic Editor

PLOS ONE